# A Rapid Self-Alignment Strategy for a Launch Vehicle on an Offshore Launching Platform

**DOI:** 10.3390/s23010339

**Published:** 2022-12-28

**Authors:** Rongjun Mu, Tengfei Zhang, Shoupeng Li

**Affiliations:** 1School of Astronautics, Harbin Institute of Technology, Harbin 150001, China; 2Institute of Modern Optics, Nankai University, Tianjin 300350, China

**Keywords:** self-alignment, anti-swaying coarse alignment, backtracking navigation, reverse Kalman filtering, attitude error compensation, cycle-index control function

## Abstract

To reduce the impact of offshore launching platform motion and swaying on the self-alignment accuracy of a launch vehicle, a rapid self-alignment strategy, which involves an optimal combination of anti-swaying coarse alignment (ASCA), backtracking navigation, and reverse Kalman filtering is proposed. During the entire alignment process, the data provided by the strapdown inertial navigation system (SINS) are stored and then applied to forward and backtrack self-alignment. This work elaborates the basic principles of coarse alignment and then analyzes the influence of ASCA time on alignment accuracy. An error model was built for the reverse fine alignment system. The coarse alignment was carried out based on the above work, then the state of the alignment system was retraced using the reverse inertial navigation solution and reverse Kalman filtering with the proposed strategy. A cycle-index control function was designed to approximate strict backtracking navigation. Finally, the attitude error was compensated for after the completion of the first and the last forward navigation. To demonstrate the effectiveness of the proposed strategy, numerical simulations were carried out in a scenario of launch vehicle motion and swaying. The proposed strategy can maximize the utilization of SINS data and hence improve the alignment accuracy and further reduce the alignment time. The results show that the fully autonomous alignment technology of the SINS can replace the complex optical aiming system and realize the determination of the initial attitude of a launch vehicle before launch.

## 1. Introduction

The initial alignment (IA) accuracy of a launch vehicle (LV) directly affects the navigation accuracy, and the initial alignment time directly affects the launch vehicle test and launch process. Due to the influence of disturbance factors, such as wind and waves at sea [1], offshore launches of launch vehicles are subject to more severe swaying than land-based launches. It is necessary to solve the self-alignment problem of the launch vehicle under motion and swaying conditions and achieve considerable accuracy in a very short launch period. The alignment is one of the most important stages of a navigation system [2]. The optical sighting or self-sighting algorithm used for initial alignment of land-based launch vehicles cannot be directly applied to the initial alignment at sea [3]. In case of land-based launch, the optical sighting equipment is fixed and the launch vehicle is slightly swayed. If the launch is on an offshore launching platform, both the optical aiming equipment and the launch vehicle are swaying at the same time, it will be impossible to calibrate. How to propose a high-precision initial alignment method suitable for sea launch in response to the characteristics of the marine motion environment is one of the technical difficulties for the launch of launch vehicles [3].

At present, optical sighting is usually used at home and abroad for azimuth reference transfer, and filter compensation or traditional transfer alignment method is used to complete the horizontal reference transfer [3]. The fundamental principle of the common method for initial alignment of sea launches of launch vehicles is shown in Figure 1. The principle of this method is simple and reliable, but the application cost is high, and the accuracy is slightly worse.

Conventional self-alignment methods are performed sequentially in chronological order, and data processing is often one-off; the data of the previous stage are not used in the later stage, and there is no storage requirement, resulting in inadequate information mining [4]. In contrast, the computing power of current navigation computers is strong enough and the capacity is large enough to save the SINS data, which can then be repeatedly processed and analyzed, providing another method of improving the accuracy and speed of an LV’s self-alignment at sea.

The coarse alignment (CA) process mainly accomplishes the coarse estimation of the unknown attitude angle [5]. Qin et al. [6] proposed a coarse alignment algorithm based on gravitational acceleration, which effectively isolated the angular swaying interference by introducing the assumption of a solidification coordinate system (CS) and tracking the attitude change in real time according to the gyro output. Bai et al. [7] demonstrated the effectiveness of the gravitational acceleration-based alignment algorithm through verification experiments. The measured angular velocity of the earth’s rotation and gravitational acceleration can be used in the coarse alignment stage. Swaying is a perturbation situation that mainly affects orientation changes [8]. With the help of the idea of decomposing the inertial coordinate system, the gravity vectors in the inertial space at different moments are not co-linear, which can be solved by the double-vector attitude determination [9,10] or multi-vector attitude determination algorithm, which isolates the swaying interference [11,12,13]. To improve the accuracy and efficiency of latitude and attitude estimation by fully utilizing inertial measurements, an anti-swaying self-alignment method based on the inertia tensor of gravity observation vector was presented in Reference [2].

Coarse alignment results can be improved by applying fine alignment (FA) at the completion of the coarse alignment phase. Fine alignment improves the accuracy of the fusion process by using external sensors or information [14,15]. Tsukerman et al. [16] derived an analytic assessment of the steady-state characteristics of the fine alignment procedure.

To improve the alignment performance by taking full advantage of all sensor data in a limited amount of time [17,18,19], a real-time information-reusing alignment strategy was proposed for rotating an inertial navigation system in Reference [20]. Li et al. [21] designed a backtracking navigation scheme for INS (Inertial Navigation System)/DVL (Doppler Velocity Log) integration. Yan et al. [22] proposed an inter-travel alignment scheme based on reverse data processing, which organically combined the initial alignment of the moving base with the position navigation through the reverse navigation algorithm, which first carried out the initial alignment of the moving base attitude and the sensor sampling data storage of the SINS (Strapdown Inertial Navigation System) gyrocompass and then used the reverse and forward dead reckoning algorithms to obtain the position information, which solved the previous problem that the position needed to be corrected using landmark points after inter-travel alignment. Liang et al. [23] applied the reverse navigation algorithm to the static base alignment of the SINS, which effectively shortened the alignment time and improved the alignment accuracy. Wang et al. [24] introduced the reverse Kalman filtering technique, which processed the collected historical data several times using the forward and reverse Kalman filtering algorithms and completed the reverse fine alignment and forward combined navigation on the basis of the coarse alignment results. In Reference [25], aiming at the problem of long alignment time for traditional methods, a forward-forward backtracking initial alignment method under swaying base conditions was proposed to achieve the purpose of rapid initial alignment, which did not consider linear vibration.

The incremental output SINS is taken as the research object. Enlightened by [19], a self-alignment strategy based on the reverse data processing technology is proposed, which uses pure mathematical analysis to complete the alignment of azimuth and horizontal directions simultaneously without the need for complex optical transmission equipment, thus achieving the purpose of shortening the alignment time and improving the alignment accuracy. The main innovations of this paper are as follows:Reverse Kalman filtering algorithm is designed, and the backtracking navigation is completed by the combination of the reverse navigation solution and reverse Kalman filtering;A cycle-index control function is designed for approximate strict backtracking navigation;In addition to the attitude correction at the initial moment (IM), attitude error compensation is completed twice during the subsequent alignment.

Compared to the existing alignment methods, firstly, using of the gravity projection along the inertial system, with reverse Kalman filter technique, could avoid the effects of disturbed accelerates and angular velocities on the SINS alignment. Secondly, attitude error compensation, to a certain extent, can achieve real-time tracking, which is the most obvious difference from the existing methods. Then, the designed cycle-index control function enables the control of the number of forward and reverse navigation solutions, thus controlling the amount of computation. Simulations and experiments show that the strategy proposed is effective. In addition, the strategy can be applied to engineering practice, as it effectively improves alignment accuracy and reduces alignment time compared with the traditional alignment methods.

## 2. Problem Description

Coordinate systems and Attitude Angles are described as follows:

Geocentric Inertial CS (i system): $o_{i}-x_{i}y_{i}z_{i}$. The origin of the CS is located at the center ($o_{i}$) of the earth. The $o_{i}x_{i}$-axis points to the vernal equinox, and the $o_{i}z_{i}$-axis is along the earth’s axis of rotation. The $o_{i}y_{i}$-axis and the $o_{i}x_{i}$-axis and $o_{i}z_{i}$-axis form a right-handed CS.

Earth CS (e system): ${ο}_{i}-x_{e}y_{e}z_{e}$. The CS is fixed to the earth, and the origin is at the center ($o_{i}$) of the earth. The ${ο}_{i}x_{e}$-axis passes through the intersection of the prime meridian and the equator, the ${ο}_{i}z_{e}$-axis passes through the north pole and the ${ο}_{i}y_{e}$-axis passes through the intersection of the meridian 90°E and the equator.

Geographic CS (g system): ${ο}_{b}-x_{g}y_{g}z_{g}$. The origin of the CS is located at the center of mass of the carrier, and the ${ο}_{b}x_{g}$-axis, ${ο}_{b}y_{g}$-axis and ${ο}_{b}z_{g}$-axis point to the east, north and up, respectively.

Navigation CS (n system): ${ο}_{n}-x_{n}y_{n}z_{n}$. The CS is the reference CS for the navigation solution, and the geographic CS is used as the navigation CS in this paper.

Calculation of Navigation CS ($n^{\prime }$ system): ${ο}_{n^{\prime }}-x_{n^{\prime }}y_{n^{\prime }}z_{n^{\prime }}$. The CS is the navigation CS established according to the navigation parameters obtained from the inertial navigation solution.

Inertial Measurement Unit (IMU) CS (s system): ${ο}_{s}-x_{s}y_{s}z_{s}$. The origin of the CS is located at the center of mass of the IMU, and the three axes point to the direction of the IMU sensitive axis. It is assumed that the body CS is rigidly connected to the IMU CS.

IMU CS at the IM ($s_{0}$ system) coincides with the IMU CS (s system) at the instance of the IA, and there is no rotation relative to inertial space.

Navigation Inertial CS at the IM ($n_{0}$ system) coincides with the navigation CS (n system) at the instance of the IA, and there is no rotation relative to inertial space.

Launch CS ($n_{t}$ system): ${ο}_{t}-x_{t}y_{t}z_{t}$. The origin of the CS is located at the center of the launch point, the ${ο}_{t}y_{t}$-axis is horizontally forward along the ballistic direction, the ${ο}_{t}z_{t}$-axis is vertically upward, the ${ο}_{t}x_{t}$-axis is perpendicular to the ballistic plane to the right and the $o_{t}y_{t}$-axis and $o_{t}z_{t}$-axis form a right-handed CS. Obviously the ${ο}_{t}y_{t}z_{t}$-plane is the ballistic plane. The angle between the ballistic plane (or ${ο}_{t}y_{t}$-axis) and the local geographic north direction is usually called the launch angle, which is recorded as $A_{0}$.

The Inertial Navigation System (IMU) is installed on the LV, and its axial definition is shown in Figure 2. When the LV is “lying down” horizontally, the three axes are along the lateral, longitudinal and vertical directions respectively, i.e., “right-front-up”. Euler angles are defined as “132”:

Pitch angle (θ): After a coordinate translation so that the origin of the navigation CS coincides with the origin of the IMU CS, the pitch angle is the angle between the projection line of the longitudinal axis (${ο}_{b}y_{b}$-axis) of the LV in the ballistic plane and the ${ο}_{n}y_{n}$-axis. The angle is positive if the longitudinal axis of the offshore platform points above the horizontal plane; otherwise, it is negative, and the value range is $\theta \in [0^{\circ },180^{\circ }]$.

Yaw angle (ψ): After a coordinate translation so that the origin of the navigation CS coincides with the origin of the IMU CS, the yaw angle is the angle between the longitudinal axis of the LV and the ballistic plane. Looking at the angle plane, if the rotation from the $οy_{n}$-axis to the $ο{y^{\prime }}_{b}$-axis is counterclockwise, the angle is positive; otherwise, it is negative, and the value range is $\psi \in [-90^{\circ },90^{\circ }]$.

Roll angle (γ): After a coordinate translation so that the origin of the navigation CS coincides with the origin of the IMU CS, the roll angle is the angle between the vertical axis of the LV and the plumb plane containing the longitudinal axis of the LV. Looking forward along the longitudinal axis from the tail of the LV, if the $οz_{b}$-axis is on the right side of the vertical plane, the angle is positive; otherwise, it is negative. The value range is $\gamma \in [-90^{\circ },90^{\circ }]$.

The conversion relationship applied throughout the article is described as follows:$C_{B}^{N}$ is the representation of the attitude of *B* system with respect to *N* system. It is equivalent to say that it is the transformation matrix from *N* system to *B* system.

The differential equation for the specific force of SINS in a launch vehicle can be described as:(1)$${\dot{\nu }}_{en}^{n}\left(t\right)=C_{b}^{n}\left(t\right)f_{sf}^{b}\left(t\right)-\left(2\omega_{ie}^{n}\left(t\right)+\omega_{en}^{n}\left(t\right)\right)\times \nu_{en}^{n}\left(t\right)+g^{n}\left(t\right)$$
where $f_{sf}^{b}\left(t\right)$ is the ideal output real-time specific force of the accelerometer; $\omega_{ie}^{n}\left(t\right)$ represents the real-time angular rate of rotation of the navigation system due to Earth’s rotation; $\omega_{en}^{n}\left(t\right)$ represents the *n* system rotation of the SINS moving near the Earth’s surface due to the curvature of the Earth’s surface; $g^{n}\left(t\right)$ represents the gravity vector in the *n* system [27].

Assuming the offshore platform is moored. Influenced by the sea state, the motion state of the launching platform can be decomposed into swaying motion and linear motion. Under such conditions, the initialization of the SINS in a launch vehicle will be obtained by the self-alignment strategy designed later.

## 3. Design of Self-Alignment Method

To realize the rapid self-alignment process of the LV launched on the offshore platform, this research established a backtracking system error model, and a reverse Kalman filtering algorithm was designed. The self-alignment process is divided into three parts: ASCA, backtracking navigation FA and forward FA.

### 3.1. Self-Alignment Strategy of LV

The designed self-alignment strategy of the LV launched on the offshore platform is shown in Figure 3. In Figure 3, ${\hat{C}}_{b,C}^{n}$ represents the initial attitude direction cosine matrix calculated at the end of the CA. ${\hat{C}}_{b,R}^{n}$ represents the initial attitude direction cosine matrix calculated at the end of the backtracking FA. ${\hat{C}}_{b}^{n}(k)$ represents the initial attitude direction cosine matrix obtained from each backtracking FA estimation.

Firstly, at the IM $t_{0}$ of the alignment, the CA process is started by the ASCA method while storing IMU data in the mooring condition. The CA process is completed at the moment $t_{1}$. The computed attitude matrix ${\hat{C}}_{b,C}^{n}$, which is obtained by the CA, is used as the initial value of the first backtracking FA, and the backtracking FA uses the collected IMU data information to return the state of the LV to the IM $t_{0}$. At the same time, the attitude information estimated by the BN is combined with the known velocity information at the starting point to correct the inertial navigation. The first backtracking FA process is completed, and the attitude information with a certain accuracy is obtained. Immediately after the fast forward navigation to return the system state to the $t_{1}$ moments, the attitude errors are compensated using the data stored from $t_{1}$ to $t_{2}$ moment. Then, the backtracking-forward cyclic FA starts from moment $t_{2}$. When the last forward FA returns to moment $t_{2}$, the data stored after moment $t_{2}$ are used to catch up with the normal real-time computation until the end of the alignment. The navigation computer maintains high speed computing during the whole process. The entire FA process is completed, and higher accuracy attitude information can be obtained.

The BN algorithm is formally identical to the forward navigation algorithm. As long as the forward saved gyro data and the angular rate of Earth rotation, etc., are inverted and the forward final value is set to the backtracking initial value, BN can be realized. During the backtracking solution, all sensor data must be stored. Therefore, in the IA, if the data processed by the backtracking continues to be used as a new set of data, or if the forward and backtracking attitude solutions are combined and the data are used repeatedly, this is equivalent to increasing the quantity of data.

### 3.2. Anti-Swaying Coarse Alignment Algorithm

Through the idea of decomposition of inertial solidification CS, the initial attitude matrix $C_{s}^{n}$ is decomposed into
(2)$$C_{s}^{n}(t)=C_{n_{0}}^{n}(t)C_{s_{0}}^{n_{0}}C_{s}^{s_{0}}(t)$$

$C_{n_{0}}^{n}$ represents the transformation matrix from the navigation inertial CS ($n_{0}$-system) at the IM to the navigation CS (n-system); $C_{s_{0}}^{n_{0}}$ represents the transformation matrix from the IMU CS ($s_{0}$-system) at the IM to the navigation inertial CS ($n_{0}$-system) at the IM; $C_{s}^{s_{0}}$ represents the transformation matrix from the IMU CS (s-system) to the IMU CS ($s_{0}$ system) at the IM.

According to the angular velocity of the earth’s rotation and the time interval between the two moments, the relationship between the two moments of the navigation CS can be obtained; according to the projection of the gravitational acceleration vector at the two moments in the navigation CS at the IM of alignment, and the projection of the specific force measurement value in the initial IMU CS, the conversion relationship between the IMU CS and the navigation CS at the IM can be solved by the double-vector attitude determination algorithm. From the projection and conversion relationship represented by the above three matrices, the conversion relationship between the IMU CS and the navigation CS at the end of the alignment can be determined, and then the CA is completed.

### 3.3. Backtracking Navigation Algorithm

In order to realize the sampling and utilization of the output signals of the gyroscope and accelerometer without an omission in time, the system collects the angular incremental output of the gyroscope and the velocity incremental output of the accelerometer within a defined time interval, rather than the angular velocity output and the specific force output at discrete time points. The attitude, velocity and position update equations of the incremental SINS inertial guide can be generally expressed as follows.

#### 3.3.1. Forward Navigation [27]


(1)Attitude update


(3)$$C_{bf(m)}^{nf(m)}=C_{nf(m-1)}^{nf(m)}C_{bf(m-1)}^{nf(m-1)}C_{bf(m)}^{bf(m-1)}$$(4)$$C_{nf(m-1)}^{nf(m)}=M_{RV}^{T}(T_{m}\omega_{in(m-1/2)}^{nf}),\;\;C_{bf(m)}^{bf(m-1)}=M_{RV}(\phi_{ib(m)}^{bf})$$
where $C_{bf(m-1)}^{nf(m-1)}$ and $C_{bf(m)}^{nf(m)}$ represent the SINS attitude matrix at $t_{m-1}$ and $t_{m}$($m=1,2,3,\ldots n_{m}$), respectively. For the navigation update period $\left[t_{m-1},t_{m}\right]$,$T_{m}=t_{m}-t_{m-1}$. $t_{(m-1/2)}=\left(t_{m}+t_{m-1}\right)/2$ denotes the midpoint moment. $\omega_{in(m-1/2)}^{nf}$ represents the rotation of the *n* system with respect to the *i* system at the moment of the midpoint in the forward navigation. The subscript (m−1/2) indicates that they are the variables at the midpoint moment. $\Delta \theta_{m1}$ denotes the angular increment in the time interval $\left[\begin{array}{cc} t_{m-1} & t_{(m-1/2)} \end{array}\right]$, and $\Delta \theta_{m2}$ denotes the angular increment in the time interval $\left[\begin{array}{cc} t_{(m-1/2)} & t_{m} \end{array}\right]$. The equivalent rotation vector can be obtained using the two-subsample conical error compensation algorithm:(5)$$\phi_{ib(m)}^{bf}=\left(\Delta \theta_{m1}+\Delta \theta_{m2}\right)+\frac{2}{3}\Delta \theta_{m1}\times \Delta \theta_{m2}$$

$M_{RV}$ can be obtained from the equivalent rotation vector ϕ (ϕ=|ϕ|):$$M_{RV}=I+\frac{\sin\phi }{\phi }\left(\phi \times \right)+\frac{1-\cos\phi }{\phi^{2}}{\left(\phi \times \right)}^{2}$$


(2)Velocity update


(6)$$v_{m}^{nf(m)}=v_{m-1}^{nf(m-1)}+\Delta v_{sf(m)}^{nf}+\Delta v_{cor/g(m)}^{nf}$$
where $\Delta v_{sf(m)}^{nf}$ and $\Delta v_{cor/g(m)}^{nf}$ are the specific force velocity increment of the navigation system and the velocity increment of the detrimental acceleration in the time period $T_{m}$, respectively. $\Delta v_{rot(m)}^{bf(m-1)}$ is the rotational error compensation amount of the speed, which is caused by the rotation change of the specific force direction in space during the solution time period; $\Delta v_{scul(m)}^{bf(m-1)}$ is the paddling error compensation amount.
$$\Delta v_{cor/g(m)}^{nf}\approx -[2\omega_{ie(m-1/2)}^{nf}+\omega_{en(m-1/2)}^{nf}]\times v_{m-1/2}^{nf}T_{m}+g_{m-1/2}^{nf}T_{m},\;\Delta v_{sf(m)}^{nf}=\left[I-\frac{T_{m}}{2}(\omega_{in(m-1/2)}^{nf}\times )\right]C_{bf(m-1)}^{nf(m-1)}\Delta v_{m}+C_{bf(m-1)}^{nf(m-1)}(\Delta v_{rot(m)}^{bf(m-1)}+\Delta v_{scul(m)}^{bf(m-1)}),\;\Delta v_{rot(m)}^{bf(m-1)}=\frac{1}{2}\Delta \theta_{m}\times \Delta v_{m},\;\Delta v_{scul(m)}^{bf(m-1)}=\frac{2}{3}(\Delta \theta_{m1}\times \Delta v_{m2}+\Delta v_{m1}\times \Delta \theta_{m2}).$$
where $\Delta \theta_{m}=\Delta \theta_{m1}+\Delta \theta_{m2}$, $\Delta v_{m}=\Delta v_{m1}+\Delta v_{m2}$, $\Delta v_{m1}$ denotes the velocity increment in the time interval $\left[\begin{array}{cc} t_{m-1} & t_{(m-1/2)} \end{array}\right]$, and $\Delta v_{m2}$ denotes the velocity increment in the time interval $\left[\begin{array}{cc} t_{(m-1/2)} & t_{m} \end{array}\right]$.


(3)Position update


(7)$$p_{m}=p_{m-1}+M_{pv(m-1/2)}(v_{m-1}^{nf(m-1)}+v_{m}^{nf(m)})\frac{T}{2}$$
where $M_{pv}=\left[\begin{array}{ccc} 0 & 1/R_{Mh} & 0\\ \sec L/R_{Nh} & 0 & 0\\ 0 & 0 & 1 \end{array}\right]$, $M_{pv(m-1/2)}$ can be obtained by linear extrapolation [28].

#### 3.3.2. Backtracking Navigation

To obtain the inertial navigation attitude, velocity and position update equations for the backtracking solution, Equations (3)–(7) are transformed by shifting the terms, by taking the reverse of the angular velocity of ground rotation, the angular velocity of rotation of the navigation CS relative to the earth CS, the acceleration of gravity, the angular increment of the gyro output and the velocity increment of the angular velocity meter output to obtain:


(1)Attitude update during backtracking


(8)$$C_{br(q)}^{nr(q)}=C_{nr(q-1)}^{nr(q)}C_{br(q-1)}^{nr(q-1)}C_{br(q)}^{br(q-1)}$$(9)$$C_{nr(q-1)}^{nr(q)}=M_{RV}^{}(T_{q}\omega_{in(q-1/2)}^{nr}),C_{br(q)}^{br(q-1)}=M_{RV}^{}(\phi_{ib(q)}^{br})$$(10)$$\phi_{ib(q)}^{br}=\left(\Delta \theta_{q2}+\Delta \theta_{q1}\right)+\frac{2}{3}\Delta \theta_{q1}\times \Delta \theta_{q2}$$
where $C_{br(q-1)}^{nr(q-1)}$ and $C_{br(q)}^{nr(q)}$ represent the SINS attitude matrix at $t_{q-1}$ and $t_{q}$ ($q=1,2,3,\ldots n_{q}$), respectively. For the backtracking navigation update period $\left[t_{q-1},t_{q}\right]$, $T_{q}=t_{q}-t_{q-1}$. $t_{(q-1/2)}=\left(t_{q}+t_{q-1}\right)/2$ denotes the midpoint moment. $\omega_{in(q-1/2)}^{nr}$ represents the rotation of the *n* system with respect to the *i* system at the moment of the midpoint in the backtracking navigation. The subscript (q−1/2) indicates that they are the variables at the midpoint moment. $\Delta \theta_{q1}$ denotes the angular increment in the time interval $\left[\begin{array}{cc} t_{q-1} & t_{(q-1/2)} \end{array}\right]$, and $\Delta \theta_{q2}$ denotes the angular increment in the time interval $\left[\begin{array}{cc} t_{(q-1/2)} & t_{q} \end{array}\right]$.


(2)Velocity update during backtracking


(11)$$v_{q}^{nr(q)}=v_{q-1}^{nr(q-1)}+\Delta v_{sf(q)}^{nr}+\Delta v_{cor/g(q)}^{nr}$$
where $\Delta v_{sf(m)}^{nr}$ and $\Delta v_{cor/g(m)}^{nr}$ are the specific force velocity increment of the backtracking navigation system and the velocity increment of the detrimental acceleration in the time period $T_{q}$, respectively. $\Delta v_{rot(q)}^{br(q-1)}$ is the rotational error compensation amount of the speed, which is caused by the rotation change of the specific force direction in space during the solution time period; $\Delta v_{scul(q)}^{br(q-1)}$ is the paddling error compensation amount in the backtracking navigation system.
$$\Delta v_{cor/g(q)}^{nr}\approx -(2\omega_{ie(q-1/2)}^{nr}+\omega_{en(q-1/2)}^{nr})\times v_{q-1/2}^{nr}T_{q}+g_{q-1/2}^{nr}T_{q},\;\Delta v_{sf(q)}^{nr}=\left[I+\frac{T_{q}}{2}(\omega_{in(q-1/2)}^{nr}\times )\right]C_{br(q)}^{nr(q)}\Delta v_{q}+C_{br(q)}^{nr(q)}(\Delta v_{rot(q)}^{br(q-1)}+\Delta v_{scul(q)}^{br(q-1)}),\;\Delta v_{rot(q)}^{br(q-1)}=\frac{1}{2}\Delta v_{q}\times \Delta \theta_{q},\;\Delta v_{scul(q)}^{br(q-1)}=\frac{2}{3}(\Delta v_{q1}\times \Delta \theta_{q2}+\Delta \theta_{q1}\times \Delta v_{q2}).$$
where $\Delta \theta_{q}=\Delta \theta_{q1}+\Delta \theta_{q2}$, $\Delta v_{q}=\Delta v_{q1}+\Delta v_{q2}$, $\Delta v_{q1}$ denotes the velocity increment in the time interval $\left[\begin{array}{cc} t_{q-1} & t_{(q-1/2)} \end{array}\right]$, and $\Delta v_{q2}$ denotes the velocity increment in the time interval $\left[\begin{array}{cc} t_{(q-1/2)} & t_{q} \end{array}\right]$.


(3)Position update during backtracking




(12)
$$p_{q}=p_{q-1}-M_{pv(q-1/2)}(v_{q-1}^{nr(q-1)}+v_{q}^{nr(q)})\frac{T}{2}$$



Through the above derivation, it can be seen that the attitude, velocity and position can at the same time be basically consistent during the forward and backward navigation process, the same piece of navigation data can be solved repeatedly and the known information at the starting point can be used for correction to improve the alignment accuracy.

### 3.4. Reverse Kalman Filtering

The error equation of the BN system is deduced according to the inertial navigation error equation [25].

The actual calculated attitude matrix ${\tilde{C}}_{br}^{nr}$ has a small angle of attitude error angle with the ideal attitude matrix $C_{br}^{nr}$, then ${\tilde{C}}_{br}^{nr}$ can be expressed as
(13)$${\tilde{C}}_{br}^{nr}=\left[I-\left(\phi \times \right)\right]C_{br}^{nr}$$

The calculated attitude matrix ${\tilde{C}}_{br}^{nr}$ is generally solved by the following differential equation:(14)$${\dot{\tilde{C}}}_{br}^{nr}={\tilde{C}}_{br}^{nr}\left({\tilde{\omega }}_{ib}^{br}\times \right)-\left({\tilde{\omega }}_{in}^{nr}\times \right){\tilde{C}}_{br}^{nr}$$

From the above, it is clear that the BN process requires the inversion of the gyro output angular velocity $\omega_{ib}^{bf}$ and the rotation angular velocity $\omega_{in}^{nf}$ of the navigation system, i.e.,
(15)$${\tilde{\omega }}_{ib}^{br}=-\omega_{ib}^{bf}-\delta \omega_{ib}^{bf}$$
(16)$${\tilde{\omega }}_{in}^{nf}=-\omega_{in}^{nf}-\delta \omega_{in}^{nf}$$

Substituting the differentiation of Equations (15), (16) and (13) into Equation (14), we get
(17)$$\begin{array}{c} \left(-\dot{\phi }\times \right)C_{br}^{nr}+\left(I-\left(\phi \times \right)\right)\left[C_{br}^{nr}\left(-\omega_{ib}^{bf}\times \right)-\left(-\omega_{in}^{nf}\times \right)C_{br}^{nr}\right]\\ =\left(I-\phi \times \right)C_{br}^{nr}\left[(-\omega_{ib}^{bf}-\delta \omega_{ib}^{bf})\times \right]-\left[(-\omega_{in}^{nf}-\delta \omega_{in}^{nf})\times \right]\left(I-\phi \times \right)C_{br}^{nr} \end{array}$$

Multiplying both sides of Equation (17) simultaneously right by $C_{nr}^{br}$ and omitting the second-order small quantity,
(18)$$\left(\dot{\phi }\times \right)=\left[\left(\omega_{in}^{nf}\times \right)\left(\phi \times \right)-\left(\phi \times \right)\left(\omega_{in}^{nf}\times \right)\right]-\left(\delta \omega_{in}^{nf}\times \right)+C_{br}^{nr}\left(\delta \omega_{ib}^{bf}\times \right)C_{nr}^{br}$$

The inertial navigation attitude error equation for BN is obtained by simplifying Equation (18),
(19)$$\dot{\phi }=\left(\omega_{in}^{nf}\times \phi \right)-\delta \omega_{in}^{nf}+\delta \omega_{ib}^{bf}$$

According to the velocity update equation for BN, the differential equation of inertial navigation velocity for BN can be expressed as
(20)$${\dot{\nu }}^{nr}=C_{br}^{nr}\left(-f^{nf}\right)-\left[2\left(-\omega_{ie}^{nf}\right)+\left(-\omega_{en}^{nf}\right)\right]\times \nu^{nr}+\left(-g^{nf}\right)$$

The actual calculated velocity ${\tilde{\nu }}^{nr}$ is obtained by Equation (21).
(21)$${\dot{\tilde{\nu }}}^{nr}={\tilde{C}}_{br}^{nr}{\tilde{f}}^{nr}-\left(2{\tilde{\omega }}_{ie}^{nr}+{\tilde{\omega }}_{en}^{nr}\right)\times {\tilde{v}}^{nr}+{\tilde{g}}^{nr}$$
where ${\tilde{f}}^{br}=-f^{bf}-\delta f^{bf}$, ${\tilde{\omega }}_{ie}^{nr}=-\omega_{ie}^{nf}-\delta \omega_{ie}^{nf}$, ${\tilde{\omega }}_{en}^{nr}=-\omega_{en}^{nf}-\delta \omega_{en}^{nf}$, ${\tilde{g}}^{nr}=-g^{nf}-\delta g^{nf}$.

The velocity error $\delta \nu^{nr}$ can be obtained by subtracting Equations (21) and (20)
(22)$$\begin{array}{cc} \delta {\dot{\nu }}^{nr}= & \left[{\tilde{C}}_{br}^{nr}{\tilde{f}}^{nr}-C_{br}^{nr}(-f^{nf})\right]-\\ & \left\{\left(2{\tilde{\omega }}_{ie}^{nr}+{\tilde{\omega }}_{en}^{nr}\right)\times {\tilde{v}}^{nr}-\left[2(-\omega_{ie}^{nf})+(-\omega_{en}^{nf})\right]\times {\tilde{v}}^{nr}\right\}+\left[{\tilde{g}}^{nr}-(-g^{nf})\right] \end{array}$$

Substituting Equation (13) into Equation (22) and expanding it gives:(23)$$\begin{array}{cc} \delta {\dot{\nu }}^{nr}= & \left[\left(I-\phi \times \right)C_{br}^{nr}(-f^{bf}-\delta f^{bf})-C_{br}^{nr}(-f^{bf})\right]-\\ & \left\{\begin{array}{l} \left[2(-\omega_{ie}^{nf}-\delta \omega_{ie}^{nf})+(-\omega_{en}^{nf}-\delta \omega_{en}^{nf})\right]\times (v^{nr}+\delta v^{nr})-\\ \left[2(-\omega_{ie}^{nf})+(-\omega_{en}^{nf})\right]\times {\tilde{v}}^{nr} \end{array}\right\}-\delta g^{nf} \end{array}$$

The Equation (23) is simplified by omitting the second-order small quantity error and the gravity error $\delta g^{nf}$, and the velocity error equation for the BN is obtained as follows:(24)$$\delta {\dot{\nu }}^{nr}=-C_{br}^{nr}f^{bf}\times \phi -v^{nr}\times \left(2\delta \omega_{ie}^{nf}+\delta \omega_{en}^{nf}\right)+\left(2\omega_{ie}^{nf}+\omega_{en}^{nf}\right)\times \delta v^{nr}-\delta f^{bf}$$

The position update differential equation for BN can be written as:(25)$$\dot{L}=-\nu_{N}^{n}/\left(R_{M}+h\right)$$
(26)$$\dot{\lambda }=-\nu_{E}^{n}\sec L/\left(R_{N}+h\right)$$
(27)$$\dot{h}=-\nu_{U}$$

The position error equation for the BN is obtained from the position update differential equation Equations (25)–(27) as follows:(28)$$\delta \dot{L}=-\delta \nu_{N}/\left(R_{M}+h\right)+{\nu_{N}\delta h/\left(R_{M}+h\right)}^{2}$$
(29)$$\begin{array}{cc} \delta \dot{\lambda }= & -\delta \nu_{E}\sec L/\left(R_{N}+h\right)-\nu_{E}\delta L\sec L\tan L/\left(R_{N}+h\right)+\\ & {\nu_{E}\delta h\sec L/\left(R_{N}+h\right)}^{2} \end{array}$$
(30)$$\delta \dot{h}=-\delta \nu_{U}$$

Integrating the error equation of the BN derived above, the corresponding systematic error equation of the BN system is as follows:(31)$$\{\begin{array}{l} \dot{\phi }=\omega_{in}^{nf}\times \phi -\delta \omega_{in}^{nf}+C_{bf}^{nf}\varepsilon^{bf}\\ \begin{array}{c} \delta \dot{v}=-f^{nf}\times \phi -v^{nr}\times (2\delta \omega_{ie}^{nf}+\delta \omega_{en}^{nf})+\\ (2\omega_{ie}^{nf}+\omega_{en}^{nf})\times \delta v-C_{bf}^{nf}\nabla^{bf} \end{array}\\ \delta \dot{p}=M_{pv}\delta v+M_{pp}\delta p\\ {\dot{\varepsilon }}^{nf}={\left[\begin{array}{ccc} 0 & 0 & 0 \end{array}\right]}^{T}\\ {\dot{\nabla }}^{nf}={\left[\begin{array}{ccc} 0 & 0 & 0 \end{array}\right]}^{T} \end{array}$$

Define:$$M_{1}=\left[\begin{array}{ccc} 0 & 0 & 0\\ -\omega_{ie}\sin L & 0 & 0\\ \omega_{ie}\cos L & 0 & 0 \end{array}\right],M_{2}=\left[\begin{array}{ccc} 0 & -\frac{1}{R_{Mh}} & 0\\ \frac{1}{R_{Nh}} & 0 & 0\\ \frac{\tan L}{R_{Nh}} & 0 & 0 \end{array}\right],M_{3}=\left[\begin{array}{ccc} 0 & 0 & \frac{v_{N}^{n}}{R_{Mh}^{2}}\\ 0 & 0 & -\frac{v_{E}^{n}}{R_{Nh}^{2}}\\ \frac{v_{E}^{n}\sec^{2}L}{R_{Nh}} & 0 & -\frac{v_{E}^{n}\tan L}{R_{Nh}^{2}} \end{array}\right],\;\;\delta \omega_{ie}^{nf}=M_{1}\delta p,\;\delta \omega_{en}^{nf}=M_{2}\delta v+M_{3}\delta p,\;M_{aa}=(\omega_{in}^{nf}\times ),\;M_{ap}=-M_{1}-M_{3},\;M_{va}=-(f^{nf}\times ),\;M_{vv}=-v^{nr}\times M_{2}+((2\omega_{ie}^{nf}+\omega_{en}^{nf})\times ),\;M_{vp}=-v^{nr}\times (2M_{1}+M_{3}),\;M_{pv}=\left[\begin{array}{ccc} 0 & -\frac{1}{R_{Mh}} & 0\\ -\frac{\sec L}{R_{Nh}} & 0 & 0\\ 0 & 0 & -1 \end{array}\right],\;M_{pp}=\left[\begin{array}{ccc} 0 & 0 & \frac{v_{N}^{n}}{R_{Mh}^{2}}\\ -\frac{v_{E}^{n}\sec L\tan L}{R_{Nh}^{2}} & 0 & \frac{v_{E}^{n}\sec L}{R_{Nh}^{2}}\\ 0 & 0 & 0 \end{array}\right]$$
where ϕ, δν and δp represent the attitude error, velocity error and position error of the SINS in the three directions of East, North and Up, respectively; ε and ∇ are the gyroscopic drift and the accelerometer zero error, respectively.

According to Equation (31), the reverse Kalman filtering model of the navigation system can be deduced as:(32)$$\begin{array}{c} {\dot{X}}_{r}=F_{r}X+G_{r}W\\ Z_{r}=H_{r}X+V \end{array}$$
where $X_{r}$ is a 14-dimensional state variable; $Z_{r}$ is the observation vector; $F_{r}$ and $H_{r}$ represent the system matrix and observation matrix, respectively; $G_{r}$ is the noise allocation matrix; W and V represent the system noise and observation noise, respectively.

State variables:(33)$$X={\left[\begin{array}{lllll} \phi^{T} & \delta v^{T} & \delta p^{T} & {\left(\varepsilon^{b}\right)}^{T} & {\left(\nabla^{b}\right)}^{T} \end{array}\right]}^{T}$$

System matrix in reverse Kalman filtering:(34)$$F_{r}=\left[\begin{array}{lllll} M_{aa} & M_{av} & M_{ap} & C_{b}^{n} & 0_{3\times 3}\\ M_{va} & M_{vv} & M_{vp} & 0_{3\times 3} & -C_{b}^{n}\\ 0_{3\times 3} & M_{pv} & M_{pp} & 0_{3\times 3} & 0_{3\times 3}\\ 0_{6\times 3} & 0_{6\times 3} & 0_{6\times 3} & 0_{6\times 3} & 0_{6\times 3} \end{array}\right]$$

Observation vector:(35)$$Z_{r}=v_{SINS}$$
where $v_{SINS}$ denotes the velocity component obtained from the SINS solution.

Observation Matrix:(36)$$H_{r}=\left[\begin{array}{ccc} 0_{3\times 3} & I_{3\times 3} & 0_{3\times 9} \end{array}\right]$$

According to the above equations, more accurate attitude information can be obtained by retrospectively solving the system state through the backtracking Kalman filtering algorithm.

**Remark** **1.***The continuous-time reverse Kalman filter model derived above is discretized, and then the corresponding terms are replaced by*$\Delta \theta_{q}$*and*$\Delta v_{q}$.
$$\Delta \theta_{q}=\int_{t_{q-1}}^{t_{q}}\omega_{ib}^{br}(t)dt,\;\Delta v_{q}=\int_{t_{q-1}}^{t_{q}}f_{sf}^{br}(t)dt.$$

For a launch vehicle, the changes of $\omega_{ib}^{br}$ and $f_{sf}^{br}$ are small in a short period of time $\left[t_{q-1},t_{q}\right]$, so it is generally considered that the quadratic functions of $\Delta \theta_{q}$ and $\Delta v_{q}$ are slow quantities of time. The calculation error caused is generally small, and the relatively simple gradient integration method can be used to obtain
$$\Delta \theta_{q}=\int_{t_{q-1}}^{t_{q}}\omega_{ib}^{br}(t)dt\approx \left(\omega_{ib}^{br}(t_{q-1})+\omega_{ib}^{br}(t_{q})\right)\frac{T_{q}}{2}\approx \omega_{ib(q-1/2)}^{br}T_{q}\;\;\Delta v_{q}=\int_{t_{q-1}}^{t_{q}}f_{sf}^{br}(t)dt\approx \left(f_{sf}^{br}(t_{q-1})+f_{sf}^{br}(t_{q})\right)\frac{T_{q}}{2}\approx f_{sf(q-1/2)}^{br}T_{q}$$
where $\omega_{ib(q-1/2)}^{br}$ denotes the output angular velocity of the gyroscope at the moment of the midpoint $t_{(q-1/2)}$; $f_{sf(q-1/2)}^{br}$ denotes the output specific force of the accelerometer at the moment of the midpoint $t_{(q-1/2)}$. 

### 3.5. Cycle-Index Control Function

For incremental SINS, repeated forward and backtracking analysis to calculate the stored data is beneficial to improving the navigation accuracy. In this process, in order to simplify the calculation, some second-order small quantities are often omitted, or small angle approximation is adopted, so that the error of each iteration is accumulated. When the number of forward and backtracking is small, the effect of error on the results is not obvious. When the number of forward and backtracking is large and reaches a certain number of times, the error is amplified to the same multiple as the number of times due to repeated calculations, resulting in oscillation convergence or even the drift of the result not converging.

For the case where the swaying amplitude is small, the backtracking process approximates the strict reverse derivation process, and the solution is more accurate. Multiple iterations of forward and backtracking solutions can guarantee the accuracy and avoid the accumulation of calculation errors. At this point, the cycle-index control function is established as follows:(37)$$q=\left[\alpha \frac{t_{sim}}{t_{coar}}\right]$$
where the whole total sampling time is recorded as $t_{sim}$, the first backtracking time is recorded as $t_{coar}$, and α is the frequency control coefficient.

### 3.6. Attitude Error Compensation

During the self-alignment process of the LV launched on the offshore platform, when returning to the starting point of the BN, the velocity and attitude angle change due to the influence of waves, etc., so attitude error compensation is required. 

The first attitude error compensation: after obtaining the relatively attitude matrix at moment $t_{1}$, the attitude update is solved with its initial value using the SINS data stored after moment $t_{1}$. It is updated until the latest sampled data moment $t_{2}$, thereby obtaining the attitude matrix estimated at moment $t_{2}$.

The second attitude error compensation: the cyclic solution starts at the moment $t_{2}$, while the sampled data after $t_{2}$ are stored. After the last forward navigation to obtain the exact matrix at time $t_{2}$, the SINS data stored after moment $t_{2}$ are used as the initial value for attitude update solving. Until the IA ends, the latest attitude matrix is obtained.

## 4. Experiments and Results Analysis

**Remark** **2.***The self-alignment of the LV launched on OP is obviously affected by the line vibration. The oscillation in the convergence section of the estimated curve reflects the characteristics of the motion and swaying of the LV. For the influence of line vibration disturbance, considering that the period of the gravitational acceleration vector is very small with inertia, low-pass filtering and other methods can be used, but this is not within the scope of this study*.

**Remark** **3.***The orientation attitude of the LV’s self-alignment is the roll angle rather than the yaw angle*.

**Remark** **4.**
*The form of attitude matrix in the table is: [pitch, yaw, roll].*


The simulation process simulates the self-alignment process of the LV in the mooring condition. In the Xisha and towards the equatorial sea area, the swaying of the LV and the launch platform disturbed by the waves is approximately sinusoidal with equal amplitude oscillation in sea state level 4, the swaying amplitude does not exceed $2.5^{{}^{\circ}}$ and the swing period is 4s∼14s.

The hardware environment for simulation was an Intel(R) Core(TM) i5-8400 CPU with 2.80 GHz, 8.00 GB RAM and Windows 10 operating system.

The inertial sensor error parameters [29] were set as shown in Table 1. The data output rate of the inertial sensor was set to 200 Hz.

The swaying parameters [30] are shown in Table 2.

In the mooring condition, in addition to swaying, there are also lateral, longitudinal and vertical linear velocities. The linear velocity parameters [30] are shown in Table 3.

The installation height of the SINS in the LV does not exceed 60 m. Combining the swaying of the offshore launch platform with the linear velocity, the trajectory of the LV in the relative coordinate system is shown in Figure 4 and it is bounded by the scope. It has a displacement of approximately one meter, which is characteristic of the engineering application environment.

The simulation scheme is as follows:

Scheme A: The conventional method uses the first 300 s of data for ASCA, and then uses the next 360 s of data for conventional Kalman filtering FA.

Scheme B: ASCA uses the first 300 s of data, the first backtracking FA uses the first 300 s of data, the first forward FA uses the first 330 s of data, repeated backtracking and forward FA uses the first 330 s of data; and the final forward FA uses the first 360 s of data.

The time taken for scheme A and B is shown in Figure 5.

The time taken for scheme A is 660 s, while Scheme B takes 360 s. Therefore, the proposed scheme shortens the time by 45.5% compared with Scheme A.

### 4.1. Estimation of the Misalignment Angle at the First Cycle

Using the strategy proposed in this paper, a stochastic simulation was performed. Figure 6 shows the estimated curves of the misalignment angles during the first backtracking navigation FA and the first forward navigation FA. The arrow in Figure 6 indicates the estimated direction.

After using the reverse Kalman filtering back to the IM and correcting the SINS with the estimated results, most of the attitude misalignment angles are corrected, and the filter curve converges during the forward alignment process, which can guarantee the alignment accuracy.

### 4.2. Analysis of the Last Forward Navigation Alignment Results

The CA with an alignment time of 300 s and the FA with an alignment time of 360 s were studied to simulate the conventional self-alignment based on scheme A and the self-alignment strategy based on scheme B, and the results of attitude error obtained for scheme A and scheme B are shown in Figure 7.

A stochastic simulation was performed. In the conventional approach, the roll angle error enters the error band $\left[1^{\prime },10^{\prime }\right]$ after completing the alignment, while the roll angle error enters the error band $\left[0^{\prime },3^{\prime }\right]$ in the proposed method. Since the proposed method utilizes the SINS data from the CA stage in the FA, more information is obtained at the same time. The alignment time of the proposed strategy depends on the convergence time of the FA filter, and the CA no longer occupies the alignment time; the SINS data of the CA stage are used twice with less storage and calculation, which not only increases the quantity of data but also effectively shortens the alignment time.

The convergence segment of the simulation results from 300 s to 360 s was selected for analysis. The statistics in Table 4 are for the convergence segment.

As can be seen from Table 4, the convergence quality of the proposed strategy is significantly better than that of the conventional method. The proposed strategy makes use of the SINS data in the CA stage from the beginning and obtains more information in a limited time with better results; with the increase in the frequency of backtracking, there is no obvious adjustment in the alignment process of the method. Through analysis, the algorithm can achieve accuracy after one or two backtracking processes.

### 4.3. Simulation Analysis of Multiple Reverse Navigation

Under the above simulation conditions, four of the reverse processes were selected and the alignment error curves are shown in Figure 8. Clearly, the orientation attitude angle is continuously corrected as the number of reversals increases.

### 4.4. Monte Carlo Simulation Results and Discussion

The statistics of the results of 50 simulations under the alignment conditions of scheme A and scheme B are shown in Figure 9.

The statistical characterization of the converged three-axis attitude angle errors was carried out, and the corresponding characterization is shown in Table 5.

As can be seen from Table 5, the alignment accuracy of scheme B is obviously better than that of scheme A, its attitude angle error is less than 3.9′ and the average value is within 0.82′. The standard deviations of pitch error, roll error and yaw error are 0.0853′, 1.4095′ and 0.0123′, respectively. It can be seen that the stability of the convergence section of the attitude angle error in the horizontal direction is improved, and the stability of the convergence section of the roll angle error decreases.

### 4.5. The Effect of CA Time on Alignment Accuracy of the Proposed Method

At present, there are few studies on the BN that focus on the effect of CA time in alignment accuracy. Based on the proposed method in this paper, 50 Monte Carlo simulations were conducted using the CA time of 120, 240, 300 and 330 s. The statistical analysis of the alignment accuracy results is shown in Table 6.

In general, the ASCA algorithm has a relatively weak anti-interference ability within a short period of time, and if the CA time is extended appropriately, good CA results can often be achieved. However, it can be seen from Table 6 that appropriately extending the CA time and keeping the FA time unchanged does not necessarily improve the final alignment accuracy, which is related to the accumulation of errors during the backtracking.

### 4.6. Validation Test

To compare the proposed strategy with conventional optical aiming method, the laser IMU is mounted on a six-degree-of-freedom turntable, and the relationship between the IMU orientation and the frame angle of the turntable is determined using the prism on the IMU to establish the reference orientation. The update frequency of IMU’s raw data is set to 100 Hz. The bias stability of gyroscopes and accelerometers are better than 0.01°/h and 20 µg, respectively. Following the laboratory confidentiality requirements, we will not show the physical picture.

Different attitude commands were executed on the turntable to simulate the shaking conditions during the launch of an LV, and the alignment results obtained are shown in Table 7.

As can be seen from Table 7, the experimental results under various conditions are basically consistent. The proposed alignment strategy can track the sway of the SINS well, and the alignment accuracy is within 3′.

## 5. Conclusions and Outlook

In this paper, a rapid self-alignment strategy is proposed for the characteristics of the maritime motion environment, which can complete the initial alignment work without setting up the aiming equipment and can solve the problem that the traditional alignment algorithm is not applicable in the maritime initial alignment system model. The strategy can effectively improve the initial alignment accuracy and meet the requirements of a launch vehicle on an offshore launching platform. The results show that the proposed fully autonomous alignment algorithm can be used to achieve the initial alignment of the SINS before the launch of the LV. To meet the demand of low cost and fast launch of LVs, the fully autonomous alignment technology will be gradually applied to LVs to replace the optical aiming system.

## Figures and Tables

**Figure 1 sensors-23-00339-f001:**
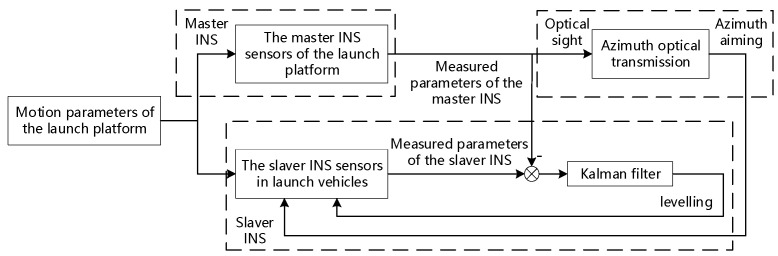
Fundamentals of initial alignment of launch vehicles.

**Figure 2 sensors-23-00339-f002:**
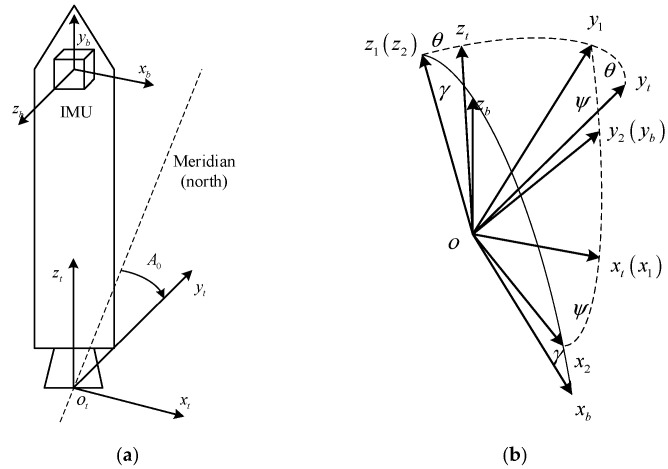
Launch CS and IMU CS of LV [26]. (**a**) Launch CS and IMU CS; (**b**) Euler angle definition.

**Figure 3 sensors-23-00339-f003:**
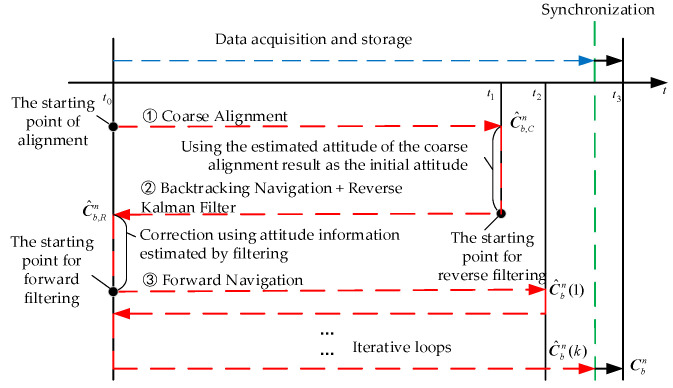
The self-alignment strategy of LV launched on the offshore platform.

**Figure 4 sensors-23-00339-f004:**
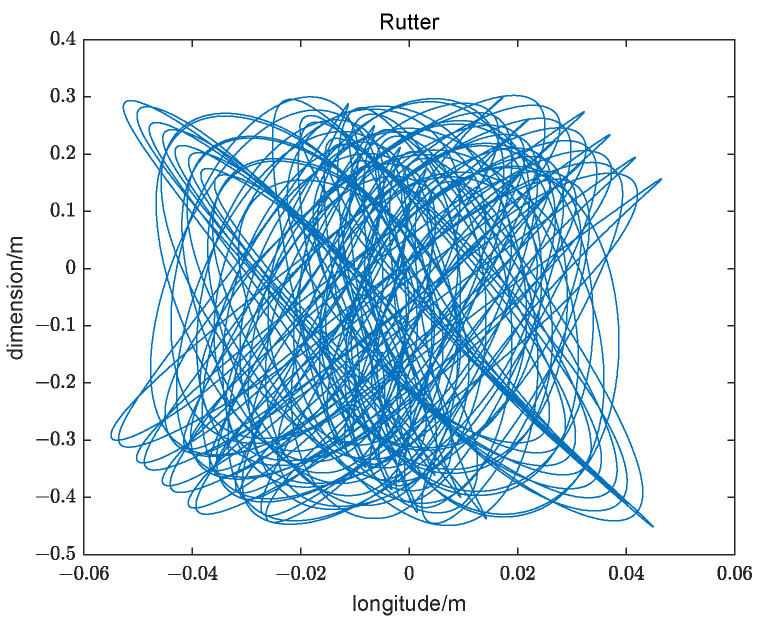
The trajectory of the LV swaying with the offshore launch platform.

**Figure 5 sensors-23-00339-f005:**
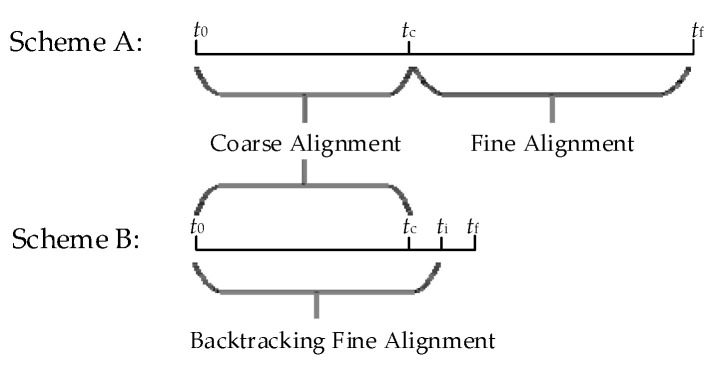
The time taken for schemes A and B.

**Figure 6 sensors-23-00339-f006:**
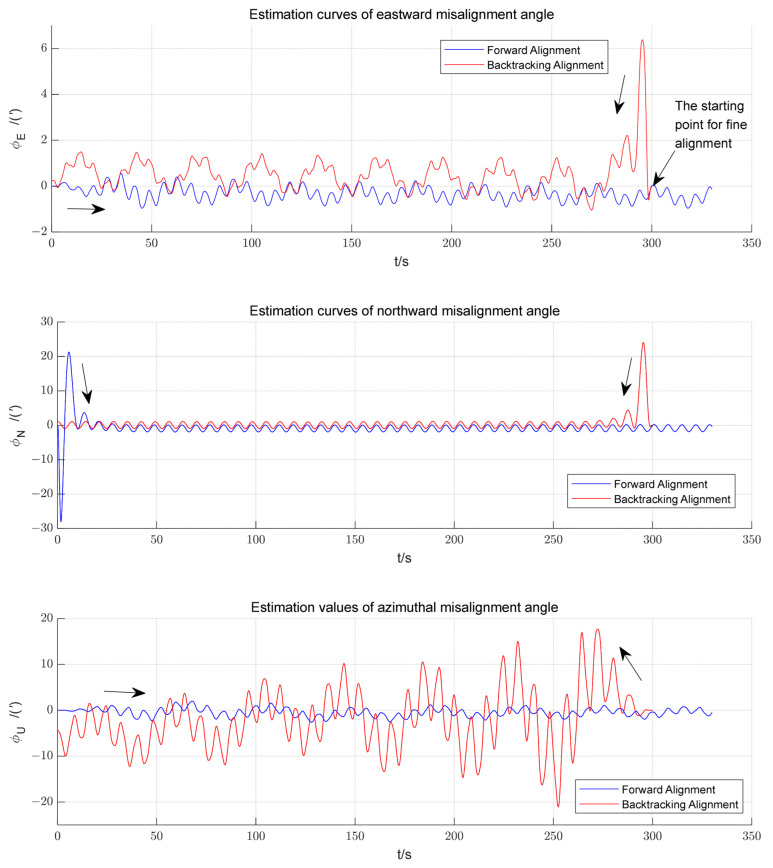
Estimated curves of the misalignment angles during the first backtracking navigation FA and the first forward navigation FA.

**Figure 7 sensors-23-00339-f007:**
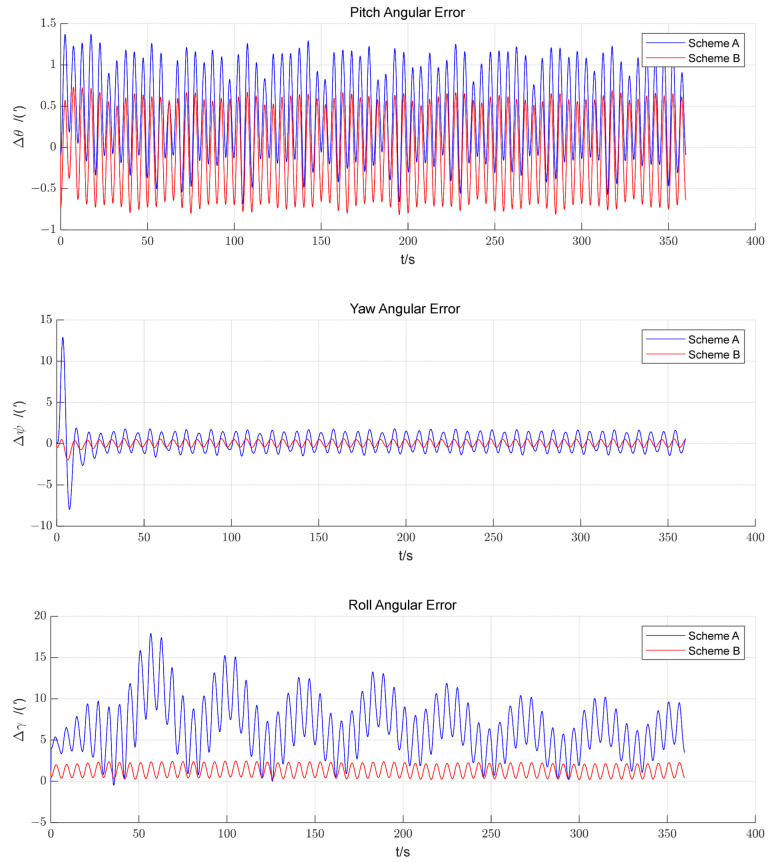
Three-axis attitude error curves for the FA of scheme A and the last FA of scheme B.

**Figure 8 sensors-23-00339-f008:**
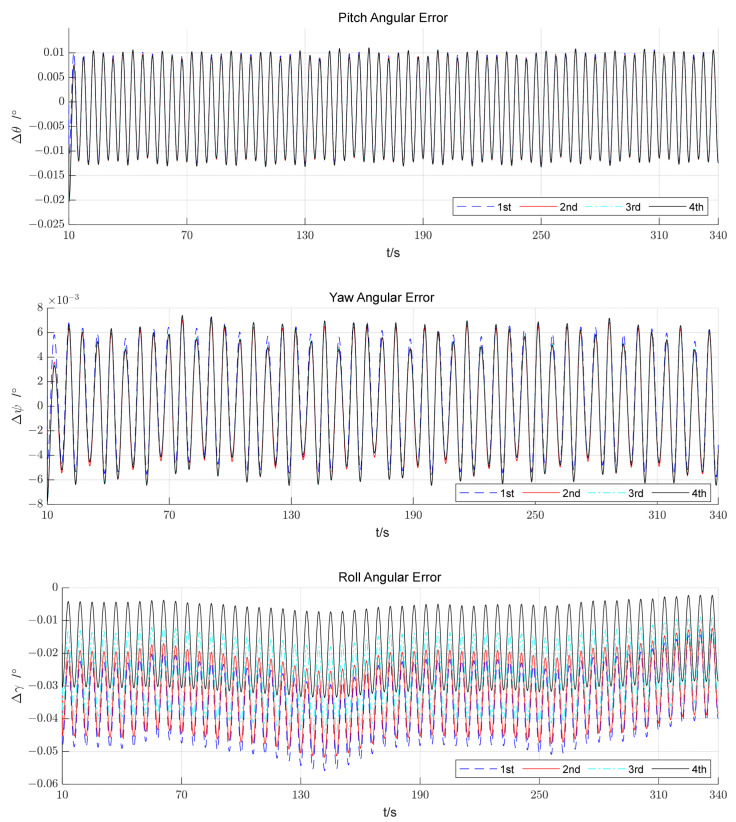
Curves of attitude errors.

**Figure 9 sensors-23-00339-f009:**
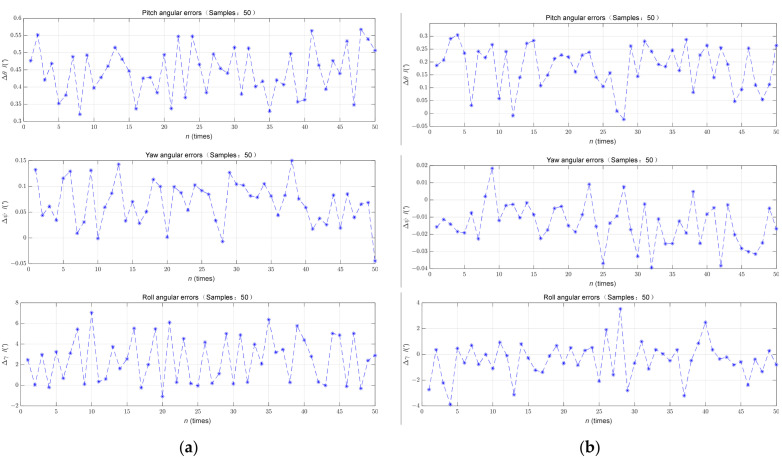
Alignment result statistics for 50 simulations: (**a**) Scheme A; (**b**) Scheme B.

**Table 1 sensors-23-00339-t001:** Sensor errors settings.

Error Terms	Gyroscope	Accelerometer
	3-Axis (1 σ)	3-Axis (1 σ)
Scale factor repeatability deviation	0.00005	0.000037
Zero bias stability deviation	0.01°/h	20 μg
Random walk coefficient	$0.005{}^{\circ}/\sqrt{h}$	$1.4\;\mu g/\sqrt{Hz}$

**Table 2 sensors-23-00339-t002:** Swaying parameter settings.

	Pitch	Roll	Yaw
Swing amplitude (°)	1.75	2.5	1.25
Swing period (s)	5	6	7

**Table 3 sensors-23-00339-t003:** Linear velocity parameter settings.

	Lateral	longitudinal	Vertical
Amplitude (m/s)	0.2	0.03	0.02
Period (s)	7	8	6

**Table 4 sensors-23-00339-t004:** Simulation results.

Scheme	Convergence Analysis			
	MEAN ( ${}^{\prime }$)	STD ( ${}^{\prime }$)/(1 σ)	MAX ( ${}^{\prime }$)	MIN ( ${}^{\prime }$)
A	[0.4063, 0.1269, 5.8172]	[0.4846, 0.9554, 2.2690]	[1.2277, 1.6750, 10.2059]	[−0.5708, −1.4288, 1.0795]
B	[−0.0415, 0.0487, 1.2252]	[0.4695, 0.3474, 0.6638]	[0.6912, 0.5822, 2.2717]	[−0.7575, −0.4713, 0.1863]

**Table 5 sensors-23-00339-t005:** Statistics for simulation result.

Scheme	Statistics			
	MEAN ( ${}^{\prime }$)	STD ( ${}^{\prime }$)/(1 σ)	MAX ( ${}^{\prime }$)	MIN ( ${}^{\prime }$)
A	[0.5065, −0.0448, 2.8971]	[0.0681, 0.0424, 2.2384]	[0.5677, 0.1502, 7.0392]	[0.3206, −0.0448, −1.0826]
B	[0.2644, −0.0169, −0.8103]	[0.0853, 0.0123, 1.4095]	[0.3048, 0.0184, 3.5274]	[−0.0229, −0.0395, −3.8959]
C	[−0.0364, 0.0928, 8.7820]	[0.0097, 0.0114, 3.9653]	[−0.0171, 0.1188, 17.0969]	[−0.0687 −0.0613, −1.6983]

**Table 6 sensors-23-00339-t006:** Statistics for simulation results.

$t_{Coarse}$	Statistics			
	MEAN ( ${}^{\prime }$)	STD ( ${}^{\prime }$)/(1 σ)	MAX ( ${}^{\prime }$)	MIN ( ${}^{\prime }$)
120 s	[0.1516, −0.0169, −0.1848]	[0.0633, 0.0145, 2.5233]	[0.2716, 0.0064, 8.5242]	[−0.0067, −0.0545, −6.3868]
240 s	[0.2075, −0.0193, 2.1955]	[0.1000, 0.0127, 3.8766]	[0.3662, 0.0087, 9.8579]	[0.0145, −0.0462, −5.4194]
300 s	[0.2644, −0.0169, −0.8103]	[0.0853, 0.0123, 1.4095]	[0.3048, 0.0184, 3.5274]	[−0.0229, −0.0395, −3.8959]
330 s	[0.1752, −0.0156, 0.3078]	[0.0728, 0.0131, 2.5647]	[0.3086, 0.0148, 6.5900]	[−0.0090, −0.0457, −7.2000]

**Table 7 sensors-23-00339-t007:** Results of highly precise alignment of laser gyroscope inertial measurement unit dynamic experiment.

No.	Turntable Position (°)			Environmental Conditions		Optical Aiming Azimuth (°)	Azimuth Error of the Proposed Strategy ( ${}^{\prime }$)
	Inner Frame	Middle Frame	Outer Frame	Angular Movement	Linear Velocity		
1	90	0.03	156.7	Amplitude: [0.35° 0.5° 0.25°] Period: [5 s 6 s 7 s]	Amplitude: [0.2 0.03 0.02] Period: [7 s 8 s 6 s]	8.986	−0.520
2	90	0.03	146.7	Amplitude: [0.35° 0.5° 0.25°] Period: [5 s 6 s 7 s]	Amplitude: [0.2 0.03 0.02] Period: [7 s 8 s 6 s]	−1.973	0.880
3	90	1.03	156.7	Amplitude: [0.35° 0.5° 0.25°] Period: [5 s 6 s 7 s]	Amplitude: [0.2 0.03 0.02] Period: [7 s 8 s 6 s]	8.986	1.790
4	90	1.03	146.7	Amplitude: [0.35° 0.5° 0.25°] Period: [5 s 6 s 7 s]	Amplitude: [0.2 0.03 0.02] Period: [7 s 8 s 6 s]	−1.973	−1.360
5	90	0.03	156.7	Amplitude: [1.75° 2.5° 1.25°] Period: [5 s 6 s 7 s]	Amplitude: [0.2 0.03 0.02] Period: [7 s 8 s 6 s]	9.772	−0.970
6	90	0.03	146.7	Amplitude: [1.75° 2.5° 1.25°] Period: [5 s 6 s 7 s]	Amplitude: [0.2 0.03 0.02] Period: [7 s 8 s 6 s]	−2.138	1.990
7	90	1.03	156.7	Amplitude: [1.75° 2.5° 1.25°] Period: [5 s 6 s 7 s]	Amplitude: [0.2 0.03 0.02] Period: [7 s 8 s 6 s]	9.772	2.170
8	90	1.03	146.7	Amplitude: [1.75° 2.5° 1.25°] Period: [5 s 6 s 7 s]	Amplitude: [0.2 0.03 0.02] Period: [7 s 8 s 6 s]	−2.138	−1.780

## Data Availability

Most of the data and models generated or used during the research appear in this publication.

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
