# Peer review of "A Rapid Self-Alignment Strategy for a Launch Vehicle on an Offshore Launching Platform"

_sensors, 2022, doi:10.3390/s23010339_

Round 1

Reviewer 1 Report

The authors proposed a novel self-alignment strategy to reduce the influence of the motions of the offshore launch vehicle on the self-alignment accuracy of a launch vehicle. The problem studied is highly concerned in engineering. The proposed strategy combines the anti-swaying coarse alignment, backtracking navigation, and reverse Kalman filtering. Simulation results show that the alignment accuracy is improved while the alignment time is reduced. However, the review has the following comments.

(1)    The abstract uses too many abbreviations that make it difficult to read. The abstract should avoid using abbreviations because it is independently used without the full paper in many situations.

(2)    The authors presented too many personal viewpoints on the background of this paper in the first three paragraph, such as “The optical sighting or self-sighting algorithm used for initial alignment of land-based launch vehicles cannot be directly applied to the initial alignment at sea”. It is not clear why they cannot be directly applied and what are the restrictions. Moreover, it would be more convincing if the authors could cite appropriate references in the first three paragraph.

(3)    The innovation point of this paper is not clear. The differences between the methods of this paper and those in the previous papers should be pointed out directly.

(4)    The authors should describe the dynamic model briefly and present the dynamic equations of the system before designing the self-alignment strategy.

(5)    How to define the stability of the convergence section?

(6)    In Line 340 Page 11, “Experiments” should be “Simulations”.

(7)    In Line 379-382 Page 13, the sentence might be duplicated.

(8)    English should be improved.

Author Response

Dear Sir/Madam,

We would like to express our thanks to you for your feedback. The paper has been rewritten, and all of your comments have been considered and addressed. Please see the attachment.

Best regards, Zhang

Reviewer 2 Report

1. Show what the abbreviations stand for in the Introduction, even though they are already in the Abstract.

2. Line 96 -104: The clear main contributions necessary to highlight the differences of the proposed method/algorithm with the existing methods.

3. Line 96 - 104: Use the present tense for explaining the authors' research results.  

4. Line 139 - 156: Range values for roll, pitch, and theta in the manuscript do not represent the range values can be represented by the 132 Euler angle type. Please explain it.

5. Notation for the rotation matrix/ transformation matrix in line 166 is not defined. Please also check all notations. Make sure all notations are defined.

6. See eq (1); definitions for the transformation matrices given in line 195-199 are incorrect. For example, C^n0_s0 is the representation of the attitude of n0 system with respect to s0 system. It is equivalent to say that it is  the transformation matrix from s0 system to n0 system. 

7. See eq(4); no clear definition of notation of many variables including the angular increments, velocity increments, etc

8. Line 310; Writing as appears in line 310 is incorrect. The integral term should be equivalent to the sum in discrete. In addition, what are the definitions of the following variables: T and q?

9. I found some typos. See line 340, for example.

10. Line 350-354; Why were heave and surge not considered?

11. Line 379-382. Please elaborate.

Author Response

(The authors gave the same response as above.)

Round 2

Reviewer 2 Report

1. Regarding the point 2, for clarity, the authors still need to highlight the differences of the proposed method/algorithm with the existing methods.

2. Regarding the point 4, for Euler Angle "132", it has singularity at pitch value = 900. So, it is not possible to cover pitch values in [0, 1800].

3. Regarding point 6, it is not only about the notation. If the authors refered to the definition in the response, then, for example, eq (2) is incorrect. Please refer to the seminal textbooks, e.g., (1) Egeland O, Gravdahl JT. Modeling and simulation for automatic control. Trondheim, Norway: Marine Cybernetics; 2002. (2) Schaub H, Junkins JL. Analytical mechanics of space systems. Aiaa; 2003

4. Regarding point 8, please add a proof that the approximation of the integral is valid. 
